# Sensing the Changes in Stratum Corneum Using Fourier Transform Infrared Microspectroscopy and Hyperspectral Data Processing

**DOI:** 10.3390/s24217054

**Published:** 2024-10-31

**Authors:** Krzysztof Banas, Agnieszka M. Banas, Giorgia Pastorin, Ngai Mun Hong, Shikhar Gupta, Katarzyna Dziedzic-Kocurek, Mark B. H. Breese

**Affiliations:** 1Singapore Synchrotron Light Source, National University of Singapore, 5 Research Link, Singapore 117603, Singapore; slshead@nus.edu.sg; 2Department of Pharmacy & Pharmaceutical Sciences, National University of Singapore, Block S4A, Level 3 18 Science Drive 4, Singapore 117543, Singapore; phapg@nus.edu.sg (G.P.);; 3P & G Singapore Innovation Center (SgIC), 70 Biopolis Street, Singapore 138547, Singapore; gupta.s.50@pg.com; 4M. Smoluchowski Institute of Physics, Jagiellonian University, Lojasiewicza 11, 30-348 Cracow, Poland; k.dziedzic-kocurek@uj.edu.pl

**Keywords:** ATR–FTIR, hyperspectral data processing, infrared microspectroscopy, intercellular lipid matrix, outlier removal, skin, stratum corneum

## Abstract

The stratum corneum (SC) forms the outermost layer of the skin, playing a critical role in preventing water loss and protecting against external biological and chemical threats. Approximately 90% of the SC consists of large, flat corneocytes, yet its barrier function primarily relies on the intercellular lipid matrix that surrounds these cells. Traditional methods for characterizing these lipids, such as Fourier transform infrared spectroscopy (FTIR), typically involve macroscopic analysis using attenuated total reflection (ATR) techniques. In this study, we introduce a novel approach for investigating SC samples at a microscopic level to gain detailed chemical insights and assess sample heterogeneity. Special emphasis is placed on advanced hyperspectral data pre-processing to ensure the accuracy and reliability of the results. We also evaluate methods for filtering out spectral data that significantly deviate from the mean and analyze the extracted mean spectra, the intensities of specific infrared peaks, and their ratios. The novelty of this work lies in its microscopic approach to analyzing the SC lipid matrix, diverging from the traditional macroscopic FTIR–ATR methods. By focusing on hyperspectral imaging and developing robust pre-processing techniques, this study provides more localized, high-resolution chemical insights. This microscopic perspective opens up the possibility of detecting subtle heterogeneities within the skin’s lipid matrix, offering deeper, previously unattainable understanding of the SC’s barrier function. Additionally, the exploration of spectral filtering methods enhances the precision of the analysis, paving the way for more refined and reliable investigations of skin structure and behavior in future research.

## 1. Introduction

The skin, as one of the largest and most complex biological membranes, plays a vital role in protecting the body from external mechanical, physical, chemical, and biological factors. Its primary function is to serve as a barrier against the harmful effects of the environment. Histologically, the skin is composed of three main layers: the epidermis, dermis, and hypodermis. The epidermis, which directly interfaces with the external environment, has a thickness of 50 to 150 μm and consists of five layers, including the stratum corneum (SC)—the outermost layer responsible for much of the skin’s barrier function.

The SC is only 10–20 μm thick and is often described using the “bricks and mortar” model that is presented in Figure 1. In this analogy, corneocytes (flattened, dead keratinocytes filled mainly with keratin (about 95%)) are the “bricks”, and the intercellular lipid matrix acts as the “mortar”. The lipid matrix is made up of ceramides (45–50% by weight), cholesterol (20–25%), free fatty acids (10–15%), and derivatives of cholesterol and glucosylceramides (15%), which are catabolic enzymes transforming the polar lipids into the non-polar compounds.

The precise composition and quantity of lipids in the SC are essential for maintaining cellular integrity and ensuring the skin’s optimal function as a selective barrier against exogenous substances. Therefore, a comprehensive and accurate characterization of the SC’s primary constituents is critical. This detailed understanding can facilitate the analysis of interactions between SC lipids and externally applied substances, providing valuable insights for the development of skincare formulations and therapeutics for dermatological applications.

Various biophysical techniques have been employed to study the molecular structure and function of the skin [1], including X-ray and neutron diffraction [2,3,4], differential scanning calorimetry [5], Raman and vibrational spectroscopy [6,7,8], electron microscopy (TEM, SEM, or Cryo-SEM) [9], nuclear magnetic resonance (NMR) [10], electron paramagnetic resonance (EPR) [11], and optical coherence tomography (OCT) [12]. Among these, Fourier transform infrared spectroscopy (FTIR) stands out as a non-invasive method for characterizing skin tissue by detecting subtle molecular changes.

Most studies employing FTIR, particularly with attenuated total reflection (ATR) accessories, primarily focus on macroscopic analyses defined by the ATR crystal size. For instance, the Golden Gate by Specac (Orpington, UK) uses crystals of 1 mm^2^ or 4 mm^2^, whereas the Specac multi-bounce features dimensions of 7.2 by 1 cm^2^, and the Smart ARK ATR–FTIR by ThermoFisher (Waltham, Massachusetts, USA) utilizes trough plates [13,14,15,16]. These macroscopic experiments are used to derive structural information about SC components and investigate molecular mechanisms involved in skin–product interactions, as well as links between skin pathology and therapeutic treatments [17,18,19].

However, none of these studies explicitly address the fact that skin samples cannot be considered uniform materials, for which a single representative spectrum suffices. Instead, the focus is typically on spectral positions, amplitudes of selected vibrations, and their ratios (e.g., symmetrical and asymmetrical CH_2_ and CH_3_ stretching). Although these parameters may offer insights into the ordering of lipid alkyl chain conformations [20], it is unclear whether these analyses are based on single spectra or averaged from multiple spectra.

Treating skin samples as homogeneous overlooks the inherent complexity and heterogeneity of the SC, potentially leading to false conclusions about its structure and interactions with applied substances.

Only few studies have explored the microscopic imaging of the SC [21,22], and those that do often present raw spectra without adequate pre-processing, making it difficult to draw definitive conclusions. Additionally, the selection of representative spectra in these studies is often arbitrary, with no clear rationale provided [16,22]. In cases where line scans of microscopic skin areas are performed, authors highlight that the spectral quality is excellent, and draw the conclusions based on only single line scan that may not capture the sample’s heterogeneity.

Given the SC’s complex structure, there is a pressing need to explore biochemical processes within its lipid matrix at a molecular level. In this study, we propose using FTIR microspectroscopy, coupled with a focal plane array (FPA) detector, to perform experiments in transmission mode. This approach allows for spatially resolved chemical analysis and enables the examination of sample heterogeneity.

A significant challenge in microspectroscopy of biological samples is their inherent heterogeneity. Although it is possible to detect the chemical composition of individual molecules in a given area, determining a representative spectrum remains a key issue. One common approach is to take the average spectrum, but this risks losing spatial information and can obscure minute differences or introduce contamination from non-uniform regions. In this paper, we compare alternative methods for outlier removal to refine the selection of representative spectra. By focusing on reliable spectrum extraction, we aim to ensure accurate analysis of molecular-level biochemical processes, as reflected in subtle spectral changes, peak shifts, or band shape variations.

To sum up, our study presents a novel microscopic approach to analyzing SC samples, focusing on hyperspectral imaging to provide detailed chemical insights and assess sample heterogeneity. Emphasis is placed on advanced data pre-processing to ensure accuracy, including methods for filtering out spectra that deviate from the mean. By moving beyond traditional macroscopic ATR–FTIR methods, this approach allows for localized, high-resolution analysis, offering deeper insights into the SC lipid matrix and its barrier function. Our analysis offers a more nuanced understanding of skin properties at a microscopic scale, which can help bridge the gap between macroscopic and microscopic phenomena and provide a foundation for future studies in this field.

## 2. Materials and Methods

### 2.1. Preparation of Isolated Stratum Corneum (SC)

Dermatomed cadaveric human skin samples (325 μm thickness) were procured from Science Care Inc., (Phoenix, AZ, USA). These skin fragments were incubated overnight at 4 °C in a 0.2 U/mL dispase solution, followed by the gentle detachment of the epidermal layer using forceps. Epidermal samples were then preserved at −80 °C for future use. Subsequently, the human epidermal membranes, with the SC facing upward, were incubated in Petri dishes over filter papers saturated with 0.1% (*w*/*v*) trypsin in a 0.5% (*w*/*v*) sodium bicarbonate solution at 37 ± 1 °C for 3 h [23]. The SC was meticulously removed, thoroughly rinsed with distilled water, and dried in a desiccator. After a 24-h drying period, the SC was briefly immersed in an acetone solution for 20 s to eliminate sebaceous lipids and dried again. The SC was then divided into small circular discs measuring 1 cm^2^ and floated over a 0.9% (*w*/*v*) sodium chloride solution containing antibacterial and antimycotic agents for 3 days. Prior to the experiment, the discs were carefully blotted dry on filter paper [24]. Samples were positioned on CaF_2_ windows (a material commonly employed in mid-IR transmission experiments due to its transparency in this range of electromagnetic radiation). Each sample was analyzed at least at three different locations. In total, 32 hyperspectral maps (scans) were acquired for the SC control groups.

### 2.2. Instrumental Configuration

All experiments were conducted at the ISMI beamline (Infrared Spectroscopy and Microscopy) of the Singapore Synchrotron Light Source (SSLS), equipped with a Vertex 80v FTIR spectrometer and a Hyperion 3000 microscope (both manufactured by Bruker Optics, Ettlingen, Germany). Transmission mode was selected as the most direct and optimal method for measuring thin sections of the SC.

ATR mode was ultimately rejected due to the potential damage caused to the sample by direct contact and the need to apply pressure to obtain any IR signal. Additionally, in the case of ATR mode, there is a requirement to clean the crystal after each experiment to avoid cross-contamination, which significantly increases the time required for the experiments.

We opted to use the IR microscope to collect representative spectra, even though many published studies in this field have used a macroscopic approach. However, due to the heterogeneity of the SC, there is significant variation in the spectra collected even for a relatively small area of 345.6 μm^2^ (the field of view covered by the FPA detector in our microscope configuration).

The FPA (focal plane array) detector, which is liquid nitrogen-cooled and has a resolution of 128 × 128 pixels^2^, was used to record the spectra in the wavenumber range of 698–3845 cm^−1^ with a spectral resolution of 8 cm^−1^. Using a 15× objective lens, the pixel size was equal to 2.7 μm^2^, providing a total area covered for each experiment of 345.6 μm^2^. Typically, 256 accumulations were co-added (resulting in a single experiment time of approximately 8 min) to improve the signal-to-noise ratio. The scanner velocity was set to 1.6 kHz. For the Fourier transform operation, the following parameters were used: a zero-filling factor of 2, a phase resolution of 32, and the Blackman-Harris 3-term function was selected for apodization.

Background scans were measured on a pure CaF_2_ window (the substrate material) and accounted for in all experiments and final spectra calculations for SC samples. These background scans were repeated frequently to ensure system and environmental stability. All spectra were recorded and are presented in this contribution in absorbance units. Absorbance has the advantage of being proportional to both the thickness and concentration of the sample and is generally the preferred quantity when applying chemometric methods to spectral datasets. The directly measured transmittance is related to the absorbance scale through a logarithmic relation. For example, 1.0 absorbance unit is equal to 10% transmittance, and 2 absorbance units are equal to 1% transmittance.

### 2.3. Software Used for Data Evaluation

All the spectral pre-processing procedures, as well as visualizations and multivariate data analysis, were performed using open-source software, namely the R environment version 4.3.2 [25] (an open-source platform for statistical computing) and RStudio version 2024.04.2 [26], utilizing the following packages: hyperSpec version 0.100.2 [27], ggplot2 version 3.5.1 [28], and Orange-Quasar software version 1.10.1 [29]. Examples of the R code used for data evaluation are provided in the Electronic Supplementary Materials (ESM) for transparency and reproducibility purposes. As mentioned earlier, in this study, 32 areas were analyzed. For each area, 16,384 (128^2^) spectra were recorded (each scan is sometimes called a hyperspectral map), resulting in a total of over 0.5 million spectra measured, processed, and analyzed.

### 2.4. Bands and Ratios Used as the Sensors of Changes in the State of Stratum Corneum

There are important bands and ratios that can be used to monitor changes in the condition of the SC. The 2920 cm^−1^ and 2850 cm^−1^ peaks in the FTIR spectra of SC correspond to the asymmetric and symmetric stretching vibrations of C-H bonds in aliphatic chains, respectively. Variations in the ratio of these peaks can signal structural and conformational alterations within the SC. The Amide I/Amide II ratio (peaks around 1652 cm^−1^ and 1548 cm^−1^, respectively) is commonly used to evaluate the secondary structure of proteins in the SC. A higher ratio generally indicates a predominance of α-helices, whereas a lower ratio suggests a higher proportion of β-sheets.

## 3. Results

### 3.1. Raw Spectra

Hyperspectral maps (spectra collected for each of the 128 × 128 pixels_2_, along with their spatial coordinates) can be presented as the intensities or integrals over specific bands of interest. This process is typically referred to as chemical mapping or imaging. One example of the measured region (visual overview shown in Figure 2a) and the total integral map (sum of intensities over the entire available spectral range) is presented in Figure 2b. The red square in Figure 2a, with dimensions of 345.6 μm in each direction, indicates the coverage area of the FPA detector in a single experiment.

Figure 3 shows an example of the calculated raw mean spectrum (red line) measured for this region, with the standard deviation represented by the semi-transparent grey ribbon. Due to factors such as the presence of potential hot or dead pixels in the FPA detector, and the non-homogeneity of the measured region (as clearly visible in Figure 2b), there are always some fluctuations in the baseline shape and intensities of each spectrum (Figure 3). Therefore, pre-processing procedures are necessary to reduce these deviations before any further analysis.

### 3.2. Comparison of Various Pre-Processing Procedures on Band Intensities, Ratios, and Dispersion

In spectral pre-processing, the first step is typically baseline correction. We applied an iterative calculation and subtraction of a polynomial line to each spectrum. We also tested the impact of performing no baseline subtraction, which is a common approach among many researchers.

For spectra normalization, we tested three methods that are widely used in the published literature:No normalization;Amide II normalization (dividing each spectrum by the intensity of the Amide II peak);Area normalization (dividing each spectrum by its mean value over the entire spectral range).

The goal of this step is to reduce system variability, primarily caused by local differences in sample thickness. The choice of pre-processing strategy can be critical to the final outcome. Figure 4 compares the integral maps (for bands at 1548, 2850, and 2920 cm^−1^) obtained from raw spectra (first row), baseline-corrected and Amide II-normalized spectra (second row), and baseline-corrected and area-normalized spectra (third row).

Additional examples of mean spectra and chemical mappings calculated for bands important for SC characterization, using all six combinations of pre-processing (with and without baseline correction, and with area, Amide II, or no normalization), are provided in the ESM file.

In this section, we focus on evaluating how the selection of pre-processing procedures influences the results. To systematically and quantitatively compare the outcomes, we applied the selected pre-processing steps to each of the 16,384 spectra in each chemical map, then calculated the mean spectrum and standard deviation, exporting the results as CSV files.

Figure 5 (top panel) shows box plots comparing the intensities of selected bands (1548, 1652, 2850, and 2920 cm^−1^) extracted from the mean spectra for the six cases: raw data (no baseline correction, no normalization), spectra with baseline correction only, and four additional cases using two normalization methods (area and Amide II) applied with and without baseline correction. All of these pre-processing scenarios are reported in the literature. Each point in the following plots (Figure 5 and Figure 6) represents the mean value, standard deviation, or ratio calculated for the 16,384 spectra measured in one region of the sample.

We demonstrate that pre-processing can be a critical parameter, especially when conclusions are based on intensities or integrals.

Figure 5 (bottom panel) presents box plots of normalized standard deviations for the intensities of the 1548, 1652, 2850, and 2920 cm^−1^ bands, calculated for each map and the six different pre-processing scenarios. Baseline correction and normalization help to reduce variability (normalized mean standard deviation) for the 1548, 2850, and 2920 cm^−1^ bands.

For example, when comparing normalized standard deviation for the 2920 cm^−1^ band, we observed a decrease from about 0.2 (for raw spectra with no correction) to 0.15 (after baseline correction only) and below 0.1 after normalization (both area and Amide II).

However, using arbitrary normalization to a specific band (Amide II, in this case) can distort experimental results, as it assumes that the concentrations of the compounds contributing to the Amide II band are identical across the measured region, which may not be a valid assumption.

It is unsurprising that the intensities of the selected absorbance peaks differ between raw and processed spectra. However, when considering band ratios, significant changes are also evident. Figure 6 shows the ratios of 1652 to 1548 cm^−1^ and 2920 to 2850 cm^−1^.

Ratios are generally less sensitive to the choice of pre-processing because dividing one band intensity by another cancels out some normalization effects. However, as Figure 6 demonstrates, baseline subtraction dramatically alters the 2920 to 2850 cm^−1^ ratio for each normalization strategy.

In summary, spectral pre-processing is a crucial step in data evaluation. Baseline correction and normalization significantly impact both band intensities and ratios. Pre-processing helps to reduce system variability, although there are usually some spectra that should be treated as outliers (those far from the mean, representative spectrum for the sample region). The development of a reliable protocol to identify and remove such spectra from the original datasets is described in the next section of this manuscript.

### 3.3. Outlier Removal

Two methods were tested to remove non-homogeneity from the datasets:Histogram analysis to remove points with extreme intensity values for selected bands;Principal component analysis (PCA), a multivariate statistical method often used for outlier detection. The results of these procedures were evaluated by assessing the changes in the dispersion of selected lines in the SC infrared spectrum, as well as by examining the size and compactness of the region identified by the algorithm as uniform.

#### 3.3.1. Threshold by Histogram Analysis

For the analysis of histograms for selected bands, the number of pixels within a particular range of intensities was plotted against these intensities. One method for identifying outliers is the use of the interquartile range (IQR), which is a measure of statistical dispersion, calculated as the difference between Q3 (the upper 75th percentile) and Q1 (the lower 25th percentile). Outliers can be defined as any observation outside the range: $\left[Q_{1}-k\left(Q_{3}-Q_{1}\right),Q_{3}+k\left(Q_{3}-Q_{1}\right)\right]$, where k is a non-negative constant.

John Tukey [30] proposed this method, where, for k = 1.5, points outside this range are considered *outliers*, and for k = 3, points are considered *far out*. Since using 1.5 IQR as a threshold for classifying outliers may appear arbitrary, we also tested two more strict values for the k parameter: k = 1 and k = 0.5. For outlier detection based on intensity histograms, we tested three prominent peaks in the SC infrared spectrum: the CH_2_ asymmetric stretch (2920 cm^−1^), the Amide I band (1652 cm^−1^), and the Amide II band (1548 cm^−1^).

In each case, we first calculated the intensity histogram for the specific band. The bin width h was automatically determined using the formula $h=2\times IQR\times n^{-1/3}$ (according to the Freedman–Diaconis rule [31], where n is the number of observations). We then computed the median value, interquartile range, and the minimal and maximal accepted intensities to classify a pixel as not an outlier. The median and interquartile range were used as measures of dispersion, as most of the histogram distributions were far from normal. Pixels classified as outliers were marked on the chemical imaging map with a different color to visualize the compactness and size of the excluded regions. An example of this analysis is shown in Figure 7. In the left column, histograms of pixel intensities for the 1548, 1652, and 2920 cm^−1^ bands are displayed, with median values marked by a red line and lower and upper thresholds marked by grey lines. The middle column shows the spatial distribution of the pixels classified as outliers (in red), whereas the right column presents the mean spectrum calculated only for the pixels classified as not outliers (black regions).

#### 3.3.2. PCA

Another method tested for reducing system variability was principal component analysis (PCA). PCA is commonly used for detecting outliers in datasets. In the first stage, principal components (PCs) are calculated as weighted linear combinations of the original variables in the dataset. The first few new variables (PCs) capture most of the variance of the system being investigated.

By calculating the distances from the median values in the new variables and setting a threshold, one can identify and mark pixels that fall outside this range. This is demonstrated in Figure 8, which shows a scatter plot of $PC_{1}$ versus $PC_{2}$. As setting the threshold is somewhat arbitrary, we tested three different threshold values to find a reasonable balance between improving the uniformity of the dataset and avoiding the removal of too many pixels.

In Figure 8, the scatter plot of $PC_{1}$ versus $PC_{2}$ scores for each spectrum is shown, with each point representing one spectrum in the new variable space (the first two dimensions: $PC_{1}$ and $PC_{2}$). Red-colored points indicate spectra classified as outliers. The pixels in the map visualization are also color-coded to show their spatial positions, allowing us to assess whether the outlier-free region is consistent and compact. Finally, the average spectrum for the region without outliers was calculated. In our analysis, scaling was applied before PCA to ensure that each original variable was considered equally important.

### 3.4. Comparison of Various Methods: Size of the Selected Region and Its Uniformity

Outlier detection is challenging, especially when the exclusion criteria need to be robust against several factors, such as sample size and data point distribution. Figure 9 summarizes the impact of various outlier removal techniques on the mean spectra calculated for 32 areas (each containing 16,384 spectra) measured from SC samples.

The best way to compare the performance of the outlier removal methods—histogram cutting and PCA-based—is to evaluate the dispersion (standard deviation or interquartile range) of the important bands relative to the full dataset.

Figure 9 shows the normalized standard deviation values for the 1548, 1652, 2850, and 2920 cm^−1^ bands using different outlier removal techniques. For each band, the PCA-based method consistently reduces variability more effectively than other methods. Other techniques perform well for specific bands used for outlier detection (e.g., 1548 cm^−1^) but are less effective for other bands.

The smallest effect is observed for the Amide I band (1652 cm^−1^), typically the highest peak in SC samples. Methods based on histogram cutting are less efficient at reducing variability, especially when spectra have not been pre-processed before outlier removal.

It is clear that pre-processing and removing spectra from non-uniform regions have a significant impact on the mean spectra calculated for each sample, including the line intensities and band ratios. This holds true even in simple cases where samples are taken from a control group. In more complex cases, such as when investigating stratum corneum samples treated with cosmetics, chemicals, or drugs, where uniformity of treatment must be considered, the issue becomes even more critical.

Of course, in the case of any biological sample, we should not expect FTIR spectra to be identical for each specimen in a given group, but the mean spectrum calculated for a relatively large area (345.6 μm^2^ in our case) should be very similar across control samples. As seen in Figure 3, the spread is large in mean spectra calculated without any outlier removal procedures. Only the combination of pre-processing (baseline correction and normalization) and outlier removal leads to smaller spreads and better uniformity. Additionally, we can evaluate not only the mean spectrum but also the standard deviations for the main bands of the SC spectrum.

Finally, the number of spectra remaining after outlier removal should be calculated to ensure the procedure is not overly strict and leaves a reasonable number of spectra out of the original 16,000+. In our case, depending on the chosen threshold level, around 80% of spectra remained for k = 1.5, 95% for k = 1.0, and 98% for k = 0.5 after outlier removal.

## 4. Discussion

This study presents an advanced approach for analyzing stratum corneum (SC) samples at the microscopic level using hyperspectral infrared imaging to achieve detailed chemical insights and assess sample heterogeneity. Traditional macroscopic methods, such as ATR–FTIR, often lead to inaccuracies due to the averaging of large areas, which can obscure spatial information and include contaminants. This research highlights the importance of selecting a proper strategy for representative spectrum extraction from hyperspectral datasets to avoid false conclusions. The main challenge addressed in this work is the risk of misinterpreting spectral data due to variations in sample thickness and morphological differences. Standard methods, which rely on either the mean spectrum or a single spectrum from a region of interest, are shown to potentially introduce artifacts. To overcome these issues, the study emphasizes the need for pre-processing steps such as baseline correction, normalization, and outlier removal using techniques like histogram cutting and principal component analysis (PCA). PCA is particularly effective for identifying and removing outliers from complex hyperspectral datasets, as it reduces dimensionality and flags spectra far from the centroid in the new variable space. The study demonstrates that proper pre-processing, including outlier removal, significantly improves the accuracy of the mean spectra, reducing variability and offering a more reliable comparison of biochemical processes. By moving beyond traditional macroscopic FTIR methods, our research provides a more localized and high-resolution view of the SC lipid matrix, enabling the detection of subtle heterogeneities within the skin’s barrier function. This microscopic approach offers deeper insights into SC structure and behavior, paving the way for more precise investigations in skin research.

This study represents preliminary work, in which a total of 32 areas were analyzed, yielding over 0.5 million spectra—a considerable dataset from an analytical standpoint. However, this dataset encompasses only 3,822,059.52 μm^2^ (or 0.0382 cm^2^), reflecting a relatively limited biological sample area. We are therefore still in the early stages of establishing definitive protocols for spectral pre-processing and outlier removal. Our primary objective in this work was to underscore the critical importance of meticulous pre-processing and rigorous outlier exclusion when employing IR microspectroscopy to assess changes within SC. These steps are essential to ensure that experimental artifacts and potential contaminants are not inadvertently included in the representative spectra.

Ultimately, this work not only enhances the methodology for spectral data analysis in SC studies but also bridges the gap between macroscopic and microscopic observations. The findings contribute to a more refined understanding of skin properties and set a foundation for future research in the field of hyperspectral imaging and skin biology.

## Figures and Tables

**Figure 1 sensors-24-07054-f001:**
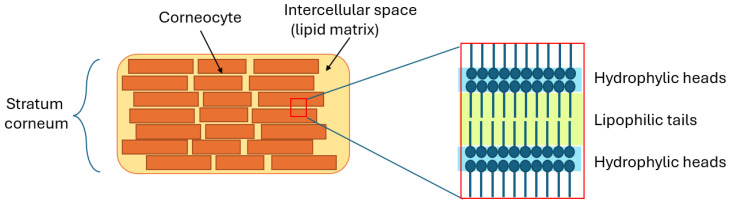
Simplified illustration of stratum corneum—“Bricks and Mortar” model.

**Figure 2 sensors-24-07054-f002:**
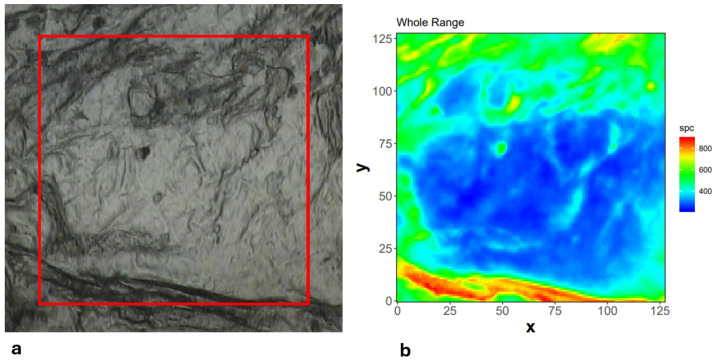
FTIR microspectroscopy experiment: (**a**) visual overview of the measured area for one experiment—red square shows the coverage of FPA detector, (**b**) integral map over all wavenumbers for the measured region in false colour scale calculated for raw spectra.

**Figure 3 sensors-24-07054-f003:**
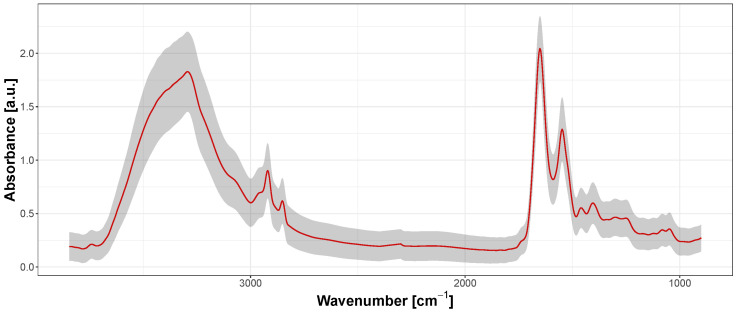
Mean raw spectrum (red line) plotted with standard deviation marked as semi-transparent grey ribbon.

**Figure 4 sensors-24-07054-f004:**
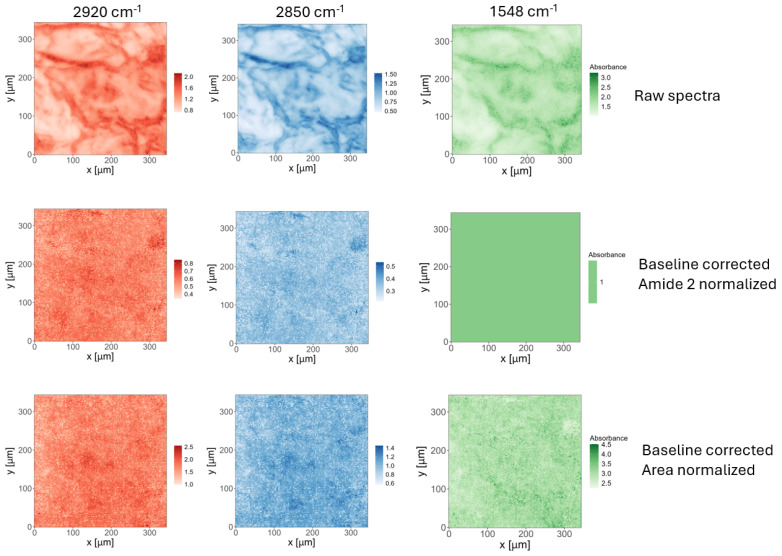
Comparison of chemical maps obtained for three different pre−processing strategies: no pre−processing, baseline correction and Amide II normalization, baseline correction and area normalization.

**Figure 5 sensors-24-07054-f005:**
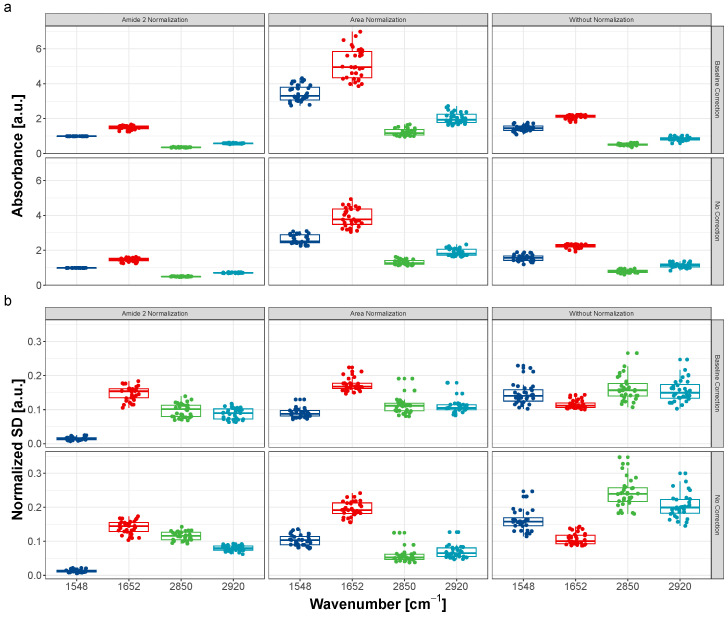
Box plots of (**a**) mean intensities (**b**) normalized SD of 1548, 1652, 2850 and 2920 cm^−1^ bands calculated for each map and six various pre-processing scenarios.

**Figure 6 sensors-24-07054-f006:**
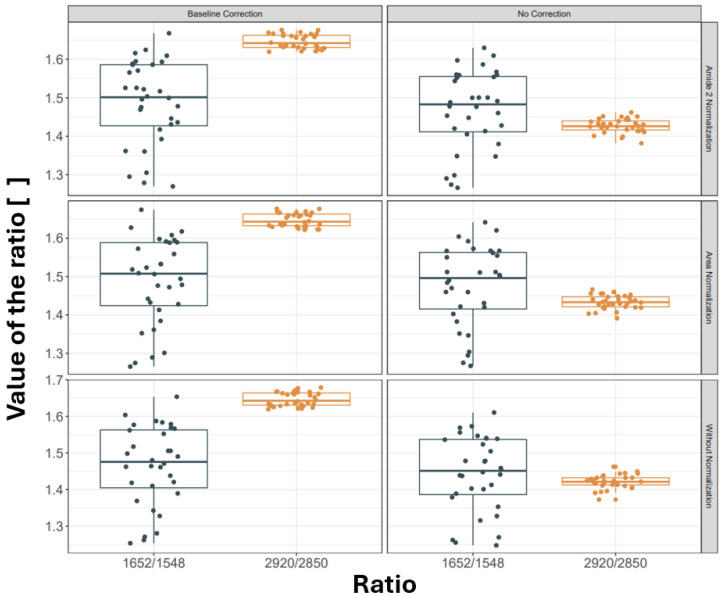
Box plots of mean values of ratios for 1652 to 1548 cm^−1^ and 2920 to 2850 cm^−1^ bands calculated for each map and six various pre-processing scenarios.

**Figure 7 sensors-24-07054-f007:**
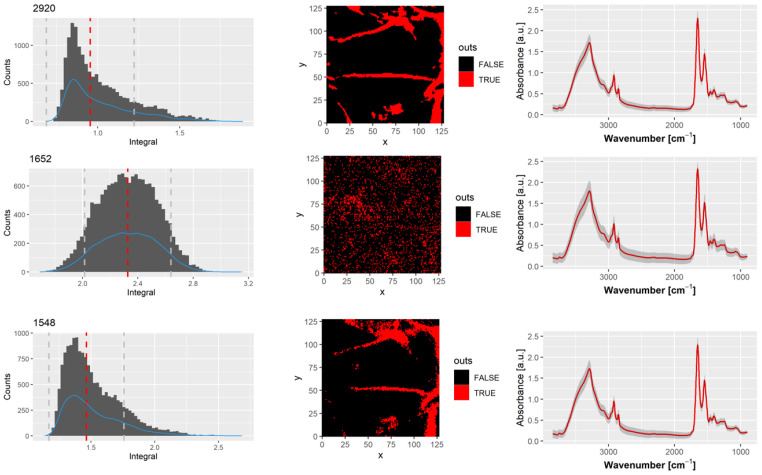
Histogram of the pixel count for the intensities of 1548, 1652 and 2920 cm^−1^ bands, maps of the location of the outliers pixels, and mean spectra with standard deviation after outliers removal.

**Figure 8 sensors-24-07054-f008:**
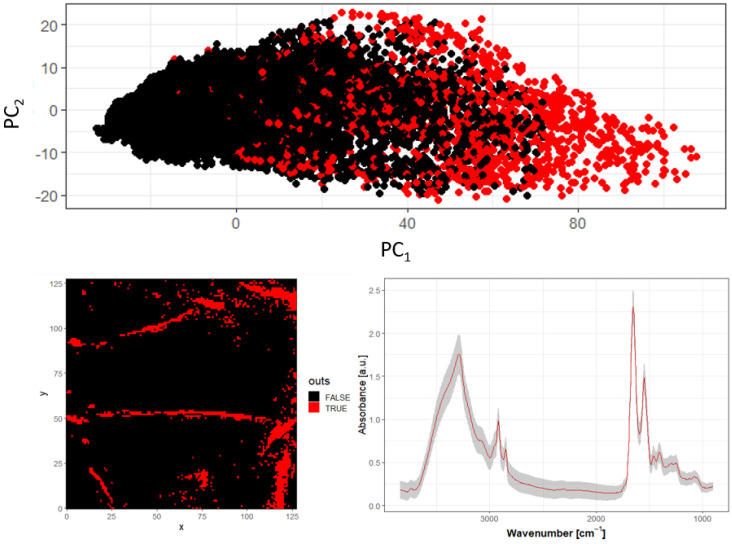
Score plot for the first two PCs with outliers marked in red, map of the location of the outliers pixels, and mean spectrum with standard deviation after outliers removal.

**Figure 9 sensors-24-07054-f009:**
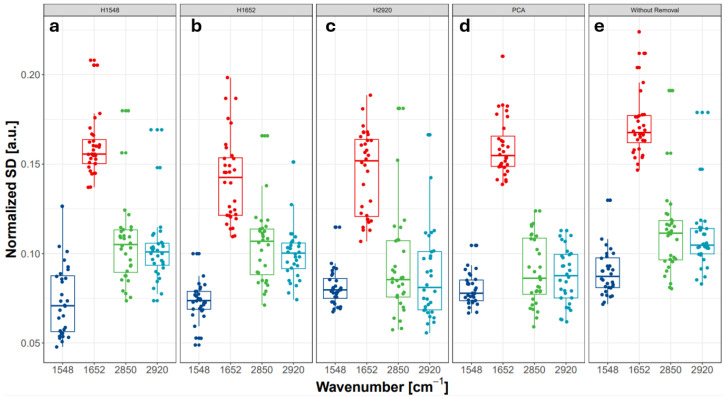
Comparison of normalized standard deviations calculated for 1548, 1652, 2850 and 2920 cm^−1^ bands for various sub-setting (outlier removal) methods.

## Data Availability

Research data used in this study are available upon request. Electronic Supplementary Material containing additional visualizations and R code used to create the plots is attached to this submission and available at MDPI.

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
