# Peer review of "Sensing the Changes in Stratum Corneum Using Fourier Transform Infrared Microspectroscopy and Hyperspectral Data Processing"

_sensors, 2024, doi:10.3390/s24217054_

Round 1

Reviewer 1 Report

Comments and Suggestions for Authors

In this work (Sensing the Changes in Stratum Corneum using Attenuated Total Reflection Fourier Transform Infrared (ATR-FTIR) Microscopy and Hyperspectral Data Processing), authors propose an approach to measure the barrier function provided the inter-cellular lipid matrix that surrounds corneocytes cells, which present 90% of the content of stratum corneum (SC), the uppermost layer of the skin that protects the body from water loss and invasion of biological and chemical agents. This approach characterizes these lipids by obtaining microscopic chemical information of these lipids to investigate their heterogeneity using ATR-FTIR Microscopy along with hyperspectral data processing. In other words, using FTIR microspectroscopy combined with Focal Plane Array (FPA) detector to perform experiments in transmission mode in order to obtain spatially resolved chemical information and to test the homogeneity or heterogeneity of the sample.

Here are my comments on the manuscript:

·       The quality of writing is poor and requires extensive improvement; thus, I suggest that a native English speaker or a professional editing service to go over the entire to enhance the quality of writing.

·       iThenticate report shows 16% similarities (report attached), I suggest that authors cut down this percentage by at least 50%. 

·       I suggest that authors rewrite the abstract, reflecting on the novelty of the present work.

·       The introduction section is crowded with no clear problem statement and also with vague and objectiveless statements; I strongly recommend authors to rewrite the ‘Introduction Section’ aiming for precise and clear analysis to what has been previously done, current problem and a proposed solution.

·       I strongly suggest that authors cite and analyze individual previously reported work that employed ARR-FTIR to characterize SC and not citing them as a group of references, as is the case in the current manuscript.

·       Although there are overwhelming number of analytical results included in this work, the manuscript lacks in-depth discussion and comprehensive analysis for the results.   

Comments on the Quality of English Language

The quality of writing is poor and requires extensive improvement; thus, I suggest that a native English speaker or a professional editing service to go over the entire to enhance the quality of writing.

Reviewer 2 Report

Comments and Suggestions for Authors

In this paper, authors propose a new way to measure stratum corneum layer of the skin by means of ATR-FTIR spectroscopy. The approach is based on the analysis of the acquired spectra to characterize the features of the stratum corneum and authors propose methods for outlier removal for the extraction of the correct parameters to be investigated.

I suggest major revision with the following comments. In particular, English language should be improved.

1)     In the authors list, remove the double comma between: Ngai Mun Hong, , Shikhar Gupta

2)     In Citation section it is mentioned:Banas, K.; Banas, A.; Pastorin, G. Sensing the Changes in Stratum Corneum using ATR-FTIR Microscopy.Sensors 2024, 1, 0.’ Please check if the authors list is correct, since it is not correspondent to the authors list.

3)     In the abstract, the phrase: ‘Extracted mean spectra, intensities of the selected IR peaks, as well as their ratios are evaluated’ contains the abbreviation IR that is not specified in the text.

4)     Keywords should be in alphabetical order, please check and correct.

5)     In the introduction, the skin is presented as a multi-layer structure: it would be interesting to add a schematic representation of the skin with every layers and thickness.

6)     In Figure 1, SR cells are represented as rectangular structure (orange in the figure). What is the yellow background? Please specify in the figure to help understanding.

7)     In 2.1, the procedure for the preparation of the stratum corneum layers is presented. How the stratum corneum was removed from the skin sample? What about the thicknesses of each sample and tolerances? Measurements were performed in 3 different location: why this choice and is there any difference for each measurement/position?

8)     In 2.2, please check the phrase: ‘After long discussion and trial experiments, transmission mode was selected as the most direct and optimal method for measuring thin sections of SC.’ English should be improved, it is not necessary to explain about long discussions/trials. I strongly suggest to review the text using a more technical approach. In general in the text, a more professional English language should be adapted.

9)     In 2.2, instead of ‘Equipment’, I suggest: ‘Instrumental configuration’. A figure of the setup used for spectra collection should be included in the text

10)  In the results section, Figure 2a and 2b should be improved, in particular Figure 2b should be enlarged in dimension. In the text, it is not clear how the figure was obtained: I suggest to add more information about data elaboration, software, process to obtain the figure. Is it a raw data? If not, it should be cited in another section or, not in ‘’Raw spectra’.

11)  Figure 3 shows an example of spectrum obtained for SC layer. We can observe different peaks but no explanation is reported. The Figure should be supported by a more complete description of it. For example, are peaks related to water absorption? Or is there any other phenomen related to the absorbance?  

12)  In 3.2, baseline correction method is reported. The figure (Figure 4) should be improved: x-y axis and values are not visible, as in Figure 5. Please improved all the figures.

13)  In Figure 5, some data present higher standard deviations and dispersion. Is there any phenomenon/parameter that affect the results? Which is the main parameters that causes dispersion?

14)  In Figure 6, which is the value measurement unit (y-axis?) It is not reported

15) For clarity, I suggest to label the graphs in Figure 9 with a) b) c) d) e) and explain in the text each result, or Improve the font and quality of the figure to make it more clear. 

Comments on the Quality of English Language

Must be improved

Reviewer 3 Report

Comments and Suggestions for Authors

 This study developed a new way of hyperspectral data processing on Attenuated Total Reflection Fourier Transform Infrared spectroscopy. The optimization of spectra and data processing, including outlier removing, mean spectra extraction, ratios of selected IR peaks were analyzed. The research design is appropriate and the methods are adequately described fluently. The results to some extent provide a new idea to sense the changes in stratum corneum and a deep understanding of the skin attributes, although there is still a big gap in the practical application. To sum up, I recommend this manuscript accept in present form.

Reviewer 4 Report

Comments and Suggestions for Authors

Dear authors,

congratulations on this work. I think it could even become a commercial technology, given the methodological deficiency in data processing, as presented!

My only observation is that in the paragraph following Figure 1, you correctly state that vibrational spectroscopy, particularly FTIR, is a non-invasive analytical technique. However, in the following paragraph, when discussing other authors who used FTIR to analyze “skin samples taken as reference,” they raise, in my opinion, doubts about these samples in relation to the non-invasive aspect. I think it would be interesting to describe the origin of these samples a little better so that doubts are not generated when reading the article, and so that the reader is not forced to look for the information in the original article, possibly referring them to another journal.

Round 2

Reviewer 2 Report

Comments and Suggestions for Authors

I still have comments before considering the manuscript suitable for publication in Sensors. In particular, I have comments regarding how precise the results were obtained since standard deviation of some group of measurements, at specific bands, are higher compared to other absorption bands.

1) I suggest to carefully review and explain the data extraction. For example, authors in the revised version explain the fact that the measurements were performed at three different locations, selected randomly to assess the homogeneity/heterogeneity of the samples.

In a comment of the revised version of the manuscript, authors justify the high standard deviation due to variations in sample thickness and other parameters. However, in a previous comment, it is declared that the thickness of the SC was similar for each sample obtained. In my opinion, it is important to identify sources of fluctuations and how to reduce these effects of fluctuations that are not induced by the quantity that the sensor need to measure. It would be important to explain which parameters strongly affect the measurement and which are negligible for the scope of the develop system.

2) In Figure 6, the measurement unit of the x-axis is missing.

3) In Figure 4, axes are not visible. It is strongly suggest to revised all the figures with a readable font.

Comments on the Quality of English Language

English is improved. Please carefully review the text of the revised manuscript since it has been modified with new contents
